# Impact of Digestive Inflammatory Environment and Genipin Crosslinking on Immunomodulatory Capacity of Injectable Musculoskeletal Tissue Scaffold

**DOI:** 10.3390/ijms22031134

**Published:** 2021-01-24

**Authors:** Colin Shortridge, Ehsan Akbari Fakhrabadi, Leah M. Wuescher, Randall G. Worth, Matthew W. Liberatore, Eda Yildirim-Ayan

**Affiliations:** 1Department of Bioengineering, College of Engineering, University of Toledo, Toledo, OH 43606, USA; colin.shortridge@rockets.utoledo.edu; 2Department of Chemical Engineering, College of Engineering, University of Toledo, Toledo, OH 43606, USA; ehsan.akbarifakhrabadi@rockets.utoledo.edu (E.A.F.); matthew.liberatore@utoledo.edu (M.W.L.); 3Department of Medical Microbiology and Immunology, University of Toledo, Toledo, OH 43614, USA; Leah.Wuescher@utoledo.edu (L.M.W.); randall.worth@utoledo.edu (R.G.W.); 4Department of Orthopaedic Surgery, University of Toledo Medical Center, Toledo, OH 43614, USA

**Keywords:** immunomodulation, cytokine, IL-4, macrophage, differentiation, inflammation, musculoskeletal, genipin, collagen, polycaprolactone, scaffold, injectable

## Abstract

The paracrine and autocrine processes of the host response play an integral role in the success of scaffold-based tissue regeneration. Recently, the immunomodulatory scaffolds have received huge attention for modulating inflammation around the host tissue through releasing anti-inflammatory cytokine. However, controlling the inflammation and providing a sustained release of anti-inflammatory cytokine from the scaffold in the digestive inflammatory environment are predicated upon a comprehensive understanding of three fundamental questions. (1) How does the release rate of cytokine from the scaffold change in the digestive inflammatory environment? (2) Can we prevent the premature scaffold degradation and burst release of the loaded cytokine in the digestive inflammatory environment? (3) How does the scaffold degradation prevention technique affect the immunomodulatory capacity of the scaffold? This study investigated the impacts of the digestive inflammatory environment on scaffold degradation and how pre-mature degradation can be prevented using genipin crosslinking and how genipin crosslinking affects the interleukin-4 (IL-4) release from the scaffold and differentiation of naïve macrophages (M0). Our results demonstrated that the digestive inflammatory environment (DIE) attenuates protein retention within the scaffold. Over 14 days, the encapsulated protein released 46% more in DIE than in phosphate buffer saline (PBS), which was improved through genipin crosslinking. We have identified the 0.5 (*w*/*v*) genipin concentration as an optimal concentration for improved IL-4 released from the scaffold, cell viability, mechanical strength, and scaffold porosity, and immunomodulation studies. The IL-4 released from the injectable scaffold could differentiate naïve macrophages to an anti-inflammatory (M2) lineage; however, upon genipin crosslinking, the immunomodulatory capacity of the scaffold diminished significantly, and pro-inflammatory markers were expressed dominantly.

## 1. Introduction

Upon injury, the inflammatory environment around musculoskeletal tissue plays a pivotal role in the success of physical and surgical therapies [1,2]. Non-steroidal anti-inflammatory drugs (NSAIDs) are commonly used to reduce inflammation, but they are also associated with the high risk of spontaneous post-injection tissue ruptures [3,4,5]. To this end, immunomodulatory tissue scaffold-based strategies have become promising approaches to control local inflammation and enhance tissue regeneration at the injury site. Immunomodulatory tissue scaffolds are designed to polarize innate immune cells toward anti-inflammatory lineages and subsequently suppress inflammation through releasing anti-inflammatory cytokines, including interleukin (IL)-10, IL-4, IL-13, and TGF-β. Among these cytokines, IL-4 is widely studied, as it polarizes macrophages into an anti-inflammatory phenotype and inhibits LPS-induced pro-inflammatory cytokine synthesis [6,7,8,9]. Several prominent studies incorporated IL-4 into the scaffold and relied on its controlled release to modulate inflammation [10,11,12,13,14]. Yet, these studies and many others in the literature largely ignored the inflammatory environment at the injury site and its deleterious effects on immunomodulatory scaffolds, including the scaffold degradation rate and cytokine release profile.

In general, the studies investigated the release characteristics of anti-inflammatory cytokines either in cell culture media, in phosphate buffer saline (PBS), or within simulated body fluid conditions [15,16]. However, upon implantation, the tissue scaffolds would be exposed to local milieus dominated by degrading enzymes in higher concentrations than the baseline levels. Clinical studies demonstrated that upon injury of musculoskeletal tissues, the level of tissue degradation enzyme matrix metalloproteinase-1 (MMP-1) increased almost 48-fold at the injury site [17,18,19,20]. Thus, a scaffold’s ability to withstand enzymatic degradation at the concentrations seen at wound sites, while preserving the sustained release rate of encapsulated anti-inflammatory cytokines, becomes paramount in immunomodulatory scaffold design strategies. 

Collagen-based scaffolds in particular garner a special interest within the musculoskeletal tissue engineering community [21,22,23,24], given that collagen proteins are the most abundant fibrous proteins within the interstitial ECM and constitute up to 30% of the total protein mass of a multicellular animal [25,26,27,28]. These features, along with its relatively low immunogenicity compared to other proteins [29], and its injectability, make collagen-based scaffolds ideal for immunomodulatory tissue scaffold material. However, limitations exist. One of the disadvantages to using collagen as an immunomodulatory material is its high susceptibility to degradation in response to multiple collagen targeting enzymes including MMP-1, MMP-2, and MMP-4 [30]. The abundance of degradation enzymes at the injury site and a subsequent burst release of incorporated anti-inflammatory cytokines remains the greatest concerns for collagen-based immunomodulatory tissue scaffolds. 

Crosslinking of the immunomodulatory scaffold may resolve the burst release of encapsulated anti-inflammatory cytokines and increase the mechanical and structural stability of scaffolds under a digestive inflammatory environment. As chemical agents, formaldehyde, glutaraldehyde, 1-ethyl-3-(3-dimethylaminopropyl) carbodiimide hydrochloride (EDC), and EDC in conjunction with N-hydroxysuccinimide (NHS) have been used to crosslink tissue scaffolds for enhanced structural stability [31,32,33,34,35]. However, these chemicals are toxic to cells and denature the anti-inflammatory cytokines incorporated into the scaffold [36,37,38,39,40]. Genipin (GP) is an alternative to currently used chemical- and physical-based collagen crosslinking agents. GP is a naturally occurring organic crosslinking agent derived from gardenia plant (Gardenia jasminoides), which has been used in Chinese medicine for its anti-carcinogenic, anti-angiogenic, and neuroprotective properties [41,42,43,44]. Compared to other chemical crosslinking agents, GP has many advantages including being a natural crosslinking agent, the stability and biocompatibility of the crosslinked products, and low toxicity [45,46,47]. Furthermore, stable crosslinked products upon GP crosslinking demonstrated resistance against enzymatic degradation [48]. GP has some limitations as well, such as few available sources and difficulties with its extraction, which lead to high cost. In addition, the strong blue pigment forms during the reaction with proteins, limiting the use of genipin [49]. Overall, genipin could be a great candidate to eliminate the destructive effects of degradation enzymes on the immunomodulatory tissue scaffold. However, the effect of GP crosslinking on the immunomodulatory capacity of scaffolds is not well studied. 

To this end, the objective of this study was three-fold: (1) to understand how the digestive inflammatory environment affects the degradation rate of the collagen scaffold; (2) whether genipin crosslinking attenuates the negative effects of the digestive inflammatory environment; and (3) to identify how genipin crosslinking alters the immunomodulatory capacity of scaffolds through IL-4 release and the promotion of anti-inflammatory differentiation of macrophages. In this study, our well-studied and characterized injectable musculoskeletal tissue scaffold (PNCOL) [50,51,52,53,54,55,56] were utilized to accomplish our objectives and to create an immunomodulatory scaffold, i-PNCOL, and genipin crosslinked (gi-PNCOL) scaffolds. 

## 2. Results

### 2.1. Bovine Serum Albumin Diminishes Scaffold Degradation in Digestive Inflammatory Environment 

Prior to running comprehensive studies with the injectable tissue scaffold, we investigated how bovine serum albumin (BSA) incorporation within the scaffold affected the scaffold degradation rate over time in the digestive inflammatory environment. Scaffolds were synthesized with and without BSA and cultured in the digestive inflammatory environment for 18 h. Kinetic collagen degradation was tracked for 14 days and demonstrated in Figure 1. The results suggest a sudden degradation for both groups on day 1. This might be the reason for detecting unpolymerized leftover collagen fibers on the scaffold surface on day 1. However, starting on day 5, the degradation of scaffolds synthesized using BSA did not increase significantly, and on day 9, it reached a plateau. Scaffolds synthesized without BSA demonstrated increased degradation over time with a rising trend. Overall, the data showed the importance of BSA in protecting scaffold structural stability in the digestive environment. Thus, it was incorporated within the scaffolds during the synthesis process. 

### 2.2. Digestive Inflammatory Environment (DIE) Attenuates Protein Retention within Scaffold 

To investigate whether the digestive inflammatory environment affects the release rate of encapsulated protein within the scaffold and to understand whether this effect is significant, long-term (21-days) and short-term (4-h) protein release was conducted in a digestive inflammatory environment (DIE) and PBS. Figure 2A shows the short-term release data. Scaffolds incubated in DIE showed the same trends as PBS-incubated scaffolds in the first and second hours with total protein release stable at 7.3 ± 1.2%. However, by the third hour, protein released in the DIE scaffolds doubled to 14.3 ± 0.7%, and by the 4th hour, they almost tripled at 20.2 ± 1.4%; while the release percentage from scaffolds cultured in PBS demonstrated the same trend for the first four hours, the release rate was significantly less than scaffolds in DIE at the 3rd and 4th h. 

A protein release study conducted over 14 days further established and demonstrated the fact that the environment in which the protein-loaded scaffolds were incorporated influences the protein release profile. DIE diminished the protein retention capacity of the injectable scaffold significantly compared to the saline environment (Figure 2B). In both environments, there was a burst release on the very first day of culture, but then, the release profile reached a plateau over time. Over 14 days, the encapsulated protein released 46% more in DIE and PBS. Specifically, 34.4 ± 4% and 23.5 ± 3% of encapsulated protein was released from the scaffolds in the DIE and PBS environment on day 14, respectively. Overall, as expected, the cumulative protein release in DIE was significantly higher (*p* < 0.05) compared to the PBS environment at all characterization time points. Thus, the scaffold may require intervention to enhance the encapsulated protein retention in DIE. 

### 2.3. Genipin Crosslinking Improves Mechanical Strength but Diminishes the Encapsulated Cell Viability and Scaffold Porosity 

Genipin crosslinking could be a viable option to prevent burst protein release from injectable scaffolds and improve a scaffold’s protein retention capacity. However, first, ideal genipin concentration needed to be identified before conducting extensive protein release and immunomodulation studies. By assessing the improvement in scaffold’s degree of crosslinking, mechanical strength, structural changes, and cytotoxicity upon crosslinking with various genipin concentration, an ideal genipin concentration was identified. Then, how the ideal genipin concentration affected the protein release from scaffolds in the digestive inflammatory environment (DIE) was assessed.

The degree of crosslinking was assessed by determining the number of free amino groups within the scaffolds after crosslinking using ninhydrin assay. Figure 3 demonstrates how free amine groups within the scaffold changed with the increased genipin crosslinking concentration. As expected, the available free amine group concentration decreased with increased genipin crosslinking concentration, which proved that the crosslinking could be achieved with as little as 0.5% (*w*/*v*) genipin crosslinking concentration. 

The changes in viscoelastic properties of the injectable scaffold were assessed using oscillatory rheology. Genipin concentration within the crosslinking solution affected the elasticity of the scaffolds. For the uncrosslinked scaffold, the storage modulus was frequency dependent and increased from 1700 ± 500 Pa to 3000 ± 800 by increasing the frequency from 0.01 to 1 Hz (Figure 4). However, the storage modulus of scaffolds crosslinked with 0.5% and 1.0% (*w*/*v*) genipin showed much weaker frequency dependence. The storage modulus increased with increasing genipin concentration from 0% (*w*/*v*) to 0.5% at all frequencies between 0.01 and 1 Hz (Figure 4). At 1 Hz, the scaffolds with no crosslinking (0% (*w*/*v*) genipin) demonstrated a storage modulus of approximately 3000 ± 800 Pa, while scaffolds crosslinked with 0.5 % (*w*/*v*) genipin solution showed an increase in storage modulus to 4700 ± 200 Pa, (*p* > 0.05). For scaffolds crosslinked with 1.0% (*w*/*v*) genipin, the storage modulus at 1 Hz was 4000 ± 500 Pa, which does not show any significant difference with the scaffold crosslinked with 0.5% genipin. 

Following the confirmation of crosslinking and mechanical changes upon various genipin concentrations, the structural changes were also investigated. Figure 5A–C demonstrates how scaffolds’ fiber diameter and pore sizes change with the increased genipin concentration.

The images clearly demonstrated that the collagen fiber thickness was increased with genipin crosslinking (Figure 5A,B). There was no statistical difference in fiber diameter for uncrosslinked (0%) and 0.1% genipin crosslinked-PNCOL scaffolds. However, with genipin concentration increased to 0.5%, fiber diameter increased 50% to 0.15 μm. The dramatic change in fiber diameter was observed for 1% genipin crosslinked-PNCOL scaffolds. The average fiber diameter was increased to 0.25 μm (≈150%) compared to 0.1 μm for uncrosslinked-PNCOL scaffolds (0%). An increase in fiber diameter with increased genipin concentration was reversely reflected in the scaffold porosity (Figure 5C). Scaffold porosity was decreased from 70% for uncrosslinked-PNCOL scaffolds to 30% porosity for genipin-crosslinked scaffolds.

To investigate the toxicity of the scaffolds, we seeded C2C12 cells in the scaffold and measured dsDNA concentration at various time points. Without genipin crosslinking (0% (*w*/*v*)), cells proliferated dramatically over the 7-day culture period. Crosslinking at 0.5% genipin concentration significantly reduced the number of viable cells compared to 0% genipin. Nevertheless, the number of cells did increase more than 2-fold at day 3 and stayed constant at the 7-day time point with 0.5% GP. Overall, cell viability was low within the scaffolds crosslinked with 1% (*w*/*v*) genipin concertation (Figure 6). Together with the changes in structural properties described in Figure 5, these data suggest that the decreased cell viability could be due to the decreased porosity and increased fiber diameter within the scaffold, which further hinders nutrition–waste exchange within the scaffold. Unlike 0% and 0.5% (*w*/*v*) genipin concentration groups, the cell number within 1% (*w*/*v*) genipin-crosslinked scaffolds did not increase over the 7-day culture period. 

### 2.4. The Optimal Genipin Concentration for Kinetic Protein Release and Immunomodulation Studies 

Based on the above findings, 0.5% (*w*/*v*) genipin concentration was found to possess the desired properties to be employed as a crosslinking agent, with a favorable balance between crosslinking capacity, mechanical, structural (fiber diameter and porosity), and cytotoxicity properties. Thus, 0.5% (*w*/*v*) genipin was used for further protein kinetic release and immunomodulation studies. 

The kinetic protein release study was conducted with and without genipin-crosslinked scaffolds to understand the effect of crosslinking on burst (4-h) and long-term (14-days) encapsulated protein release profiles. The data demonstrated that in total, 7 ± 1.4% of the protein was released from genipin crosslinked scaffolds (gi-PNCOL) within the first four hours (Figure 7A). There was a rapid release for uncrosslinked scaffolds (i-PNCOL) with 10 ± 0.6% within the first hour and in total 16 ± 0.9% within the four-hour period. During the 14-day study, genipin crosslinking prevented the burst release of the encapsulated biologics from the injectable scaffolds. This trend continued for the first two days of the culture (Figure 7B). Starting from day 4, there was no statistical difference in released protein amount between crosslinked (gi-PNCOL) and uncrosslinked (i-PNCOL) scaffolds. However, it should be noted that the cumulative protein release trend for uncrosslinked scaffold (i-PNCOL) was demonstrating the upward trend, while that released from genipin crosslinked scaffolds plateaued around day 6 until the end of day 15. 

### 2.5. Interleukin-4 Release from Injectable Scaffold Alters Macrophage Polarization

The immunomodulation capacity of released IL-4 from injectable scaffolds and the effect of genipin crosslinking on immunomodulation was demonstrated with phenotypic changes in macrophages (M0). Four groups were used in the immunomodulation study: PNCOL, g-PNCOL, i-PNOCL, and gi-PNCOL. The PNCOL and genipin (g)-PNCOL groups without incorporated IL-4 were used to understand the effect of the injectable scaffold and the scaffold with genipin on macrophage polarization independently. In i-PNCOL and gi-PNCOL groups, the effect of released IL-4 from genipin-crosslinked (gi-PNCOL) and uncrosslinked (i-PNCOL) was assessed. 

Figure 8 demonstrates the gene expression of hallmark pro-inflammatory (M1) markers of COX-2, MMP-3, MIP-2, and TNF-α. The data demonstrate that IL-4 released from i-PNCOL downregulated all M1 markers. When the scaffold was crosslinked with genipin (g-PNCOL group), COX-2 and MMP-3 expression increased 9.4 ± 0.1-fold and 5.6 ± 0.6-fold, respectively. However, when IL-4 was incorporated into the genipin-crosslinked scaffolds (gi-PNCOL), the released IL-4 decreased the COX-2 and MMP-3 expressions to 2.7 ± 0.7-fold and 4.2 ± 0.7 fold, respectively. The data suggested that genipin crosslinking increased the expression of the pro-inflammatory markers, but the incorporation of IL-4 in the genipin scaffolds can partially attenuate these effects. Interestingly, the expression of MIP-2 and TNF-α decreased for i-PNCOL scaffolds, while the same gene expressions were not detected for g-PNCOL and gi-PNCOL scaffolds.

The polarization of macrophages (M0) toward an anti-inflammatory and pro-healing phenotype (M2) was assessed with the expression of anti-inflammatory markers including CCL-18, CD206, IL-10, and CD163.

IL-4 released from injectable scaffolds (i-PNCOL) significantly increased the expression of anti-inflammatory markers (Figure 9). CCL-18 expression increased 10 ± 0.6-fold, mannose receptor expression (CD-206) increased over 12 ± 0.3-fold, IL-10 expression increased 18 ± 0.7-fold, and CD163 increased 5.5 ± 0.3-fold. Interestingly, anti-inflammatory marker expression was not detected for g-PNCOL and gi-PNCOL scaffolds (Figure 9). The data strongly indicated that IL-4 released from i-PNCOL, but not genipin-treated scaffolds, was able to differentiate macrophages (M0) to anti-inflammatory lineages; therefore, it was able to promote healing. 

## 3. Discussion

In recent years, more and more materials for immunomodulatory scaffolds have been investigated to modulate the local inflammatory environment through the release of anti-inflammatory biologics to the area of injury. However, the harsh inflammatory environment around the defect site dominated with digestive matrix metalloprotease (MMPs) and phagocytic macrophages (M1) affect the immunomodulation capacity and degradation of the scaffold. Yet, not many studies considered the digestive inflammatory environment and its effects during the design and synthesis of the immunomodulatory scaffold. In this study, we have created an IL-4-conjugated injectable nanofibrous scaffold, i-PNCOL, and systematically investigated the changes in its structural, degradation, and mechanical properties under the digestive inflammatory environment and assessed its immunomodulatory capacity with and without genipin crosslinking.

Prior to conducting intensive mechanistic studies, we confirmed the importance of bovine serum albumin (BSA), which is one of the building blocks of the injectable nanofibrous tissue scaffold as a protective agent in a digestive inflammatory environment. BSA demonstrated a protective effect against proteolytic degradation of the scaffold over 14 days (Figure 1). Albumin, particularly bovine serum albumin (BSA), limits surface adsorption and the enzymatic degradation of extracellular matrix protein molecules by interfering with collagenase activity through crowding effects [57]. Furthermore, for MMP1 (collagenase) to function effectively, both soluble calcium ions and zinc ions have to be present in the environment [58,59,60,61]. It is well-known that BSA scavenges metal ions such as zinc, copper, and iron ions within the solution and prevents the activation of matrix degradation enzyme, MMP1 [62,63].

The influence of the digestive inflammatory environment (DIE) was most remarkable in terms of the kinetic protein release profile. The sustained release of anti-inflammatory cytokines in the inflammatory environment is very crucial to modulate the inflammatory response [64,65]. Thus, the immunomodulatory scaffolds should possess a controlled-release profile of encapsulated biologics in the inflammatory environment. However, in general, the kinetic release profiles of anti-inflammatory cytokines from immunomodulatory scaffolds were assessed in PBS, cell culture media, or simulated body fluid environment, which do not represent the digestive inflammatory environment in which seen in a damaged tissue area. The degradation due to inflammation may affect the release profile of encapsulated cytokines within the scaffolds. 

Understanding how the digestive inflammatory environment affects the kinetic release profile of anti-inflammatory cytokines can help us to identify the optimal initial cytokine amount that can be incorporated within the scaffold to modulate the inflammation in an injury environment over an extended period of time. This kind of understanding also prevents excessive anti-inflammatory cytokine loading within the scaffolds. In this study, we demonstrated how the culture environment utilized for kinetic protein release studies affects the retention of the encapsulated biologics. The scaffolds cultured in DIE exhibited a burst release profile of encapsulated protein in both short-term (4 h) and long-term (14 days) culture periods (Figure 2A,B). Following 14-day culture, the scaffolds cultured in DIE demonstrated 50% more released protein compared to counterparts cultured in PBS. Loaded biologics only have a limited half-life when released into the DIE due to the presence of circulating proteolytic enzymes. For instance, bFGF has been reported to have a half-life of only 3 min, while that of PDGF is 2 min, and that of IL-4 is only 1.5 min [66,67,68,69]. A long-term sustained release would provide a constant supply of active ligands to stimulate immune cell polarization toward anti-inflammatory lineages. The crosslinking of the scaffold may address the negative effect of DIE on the burst release of encapsulated anti-inflammatory cytokines. 

We next investigated genipin as a crosslinking agent to improve the retention rate of encapsulated biologics under a digestive inflammatory environment. It is known that genipin crosslinks protein, collagen, gelatin, and chitosan and forms blue particles as a result of spontaneous reaction with primary amine groups [70,71,72]. Genipin covalently binds free amine groups present in collagen fibers, forming intra-helical and inter-helical links within the standard tropocollagen fiber, and inter-microfibrillar crosslinks (crosslinks between the fibrils that form a collagen fiber) [70,73]. Even on the assumption that genipin is an effective collagen crosslinker, we still need to confirm the optimal genipin concentration for the improvement of scaffold properties and maintaining cell viability before running extensive mechanistic studies. The degree of scaffold cross-linking with increased genipin concentration was assessed through measuring the free amine groups within the scaffold before and after genipin crosslinking (Figure 3). It was demonstrated that genipin, even at the low concentration of 1% (*w*/*v*), was able to decrease the free available amine groups within the scaffolds. 

Following confirmation of genipin-induced crosslinking, we investigated how the scaffolds’ mechanical and structural properties were altered with the crosslinker concentration. The rheological analysis (Figure 4A,B) showed that storage modulus increased by almost 50% from 3000 Pa for uncrosslinked scaffolds to 4700 Pa for 0.5% (*w*/*v*) genipin-crosslinked counterparts. Further increase in the crosslinking agent concentration had a residual impact on the storage modulus of the scaffolds. We hypothesized that the morphological changes in the scaffold upon crosslinking might be the reason for the improved mechanical outcome. Investigating the scaffolds’ fiber diameter and porosity changes before and after genipin cross linking demonstrated that with the increased genipin concentration, the diameter of collagen fibers increased 50% and 100% for the samples crosslinked with 0.5% and 1% (*w*/*v*) genipin solutions, while the porosity decreased from 70% to 30% upon crosslinking (Figure 5A–C). 

On one hand, crosslinking with 0.5% (*w*/*v*) genipin likely created an interconnected, sample-spanning network that significantly enhanced the injectable scaffold’s mechanical properties. The network formation can be attributed to the free amine groups binding efficacy of genipin. Genipin can bind not only free amine groups but also the already amine-bound genipin molecules, which provides further binding sites for genipin [48,73,74,75]. Given the fact that there are about a hundred free amine groups in one tropocollagen fibril [76], genipin crosslinking could bind already crosslinked collagen fibers and create the collagen fiber bundles as seen in Figure 5A. On the other hand, a higher concentration of genipin increased crosslinks (as shown in Figure 3) that altered the local structure and fiber thickness without significantly improving the sample spanning structure established at lower crosslinker concentrations. Thus, little additional change on the macroscopic mechanical properties such as the storage modulus was observed once the crosslinked structure was established.

Our prior studies demonstrated that we could incorporate cells within collagen-based scaffolds without any cytotoxic effects [50,52,53,54,55]. Although genipin is relatively less toxic compared to other collagen crosslinking reagents such as glutaraldehyde [36,77], it must still be used at low concentrations. The cytotoxic effect of genipin was demonstrated through an inverse correlation between increasing genipin concentrations and cell proliferation (Figure 6). The number of cells within the uncrosslinked scaffolds demonstrated an almost 4.5-fold increased over 7 days, while cell number within the scaffolds crosslinked with 0.5% (*w*/*v*) genipin solution increased 2.5-fold compared to day one. In contrast, 1% (*w*/*v*) genipin solution negatively affects cell proliferation over the seven-day culturing period. We believe that this decreased cell viability is due to possible nutrient deprivation owing to the decreased porosity of the scaffold demonstrated in Figure 5C. 

Based on our observations regarding the degree of genipin crosslinking, mechanical strength, cytotoxicity, and structural analysis indicated that the optimal genipin concentration for preserving scaffold integrity without compromising the cell viability was 0.5% (*w*/*v*). Therefore, the 0.5% (*w*/*v*) genipin concentration was utilized in kinetic protein release studies under digestive inflammatory environment and macrophage polarization studies. The kinetic protein release data demonstrated that genipin crosslinking significantly improved the protein retention within the scaffolds, especially in the first four days of culture in a digestive inflammatory environment. In the first four hours of culture (burst release period), the protein retention is dramatically increased from 6% for i-PNCOL (scaffold without crosslinking) to 16% for gi-PNCOL (genipin-crosslinked scaffolds) (Figure 7A). This effect was sustained for another four days. Starting from day 4, there was no statistical difference between the protein retention of gi-PNCOL and i-PNCOL. The bonding between the microfibrils could be cleaved after 4 days of digestive inflammatory environment and caused an increased release rate of encapsulated protein. However, our gene expression analysis confirmed that the IL-4 released from the scaffold during the 4-day period is more than enough to differentiate macrophages into anti-inflammatory lineages. 

Results from gene expression analysis (Figure 8 and Figure 9) highlight the immunomodulatory role of IL-4 released from genipin-crosslinked (gi-PNCOL) and uncrosslinked (i-PNCOL) scaffolds. Macrophages were cultured with four different scaffolds, namely: PNCOL (scaffold without IL-4 encapsulation), i-PNCOL (scaffold with IL-4 encapsulation), g-PNCOL (scaffold without IL-4 encapsulation but crosslinked with genipin), and gi-PNCOL (scaffold with IL-4 encapsulation and crosslinked with genipin). The expression of pro-inflammatory (M1) markers (COX-2, MMP3, MIP-2, and TNF-α) along with anti-inflammatory (M2) markers (CCL-18, CD206, IL-10, and CD163) were analyzed.

Genipin crosslinking increased COX-2 and MMP-3 expression, which are two important pro-inflammatory markers. Macrophages exposed to genipin-crosslinked scaffolds (g-PNCOL) expressed 10-fold and 5.5-fold increases in COX-2 and MMP3 expressions, respectively (Figure 8). When macrophages cultured with IL-4 encapsulated and genipin-crosslinked (gi-PNCOL), IL-4 released from the scaffold reducde the expressions from 10-fold to 3-fold for COX-2 and from 5.5-fold to 4-fold for MMP-3. Regarding MIP-2 and TNF-α expression, IL-4 release from i-PNCOL decreased the expression of these pro-inflammatory markers compared to PNCOL; however, no MIP-2 and TNF-α expression was detected for genipin-crosslinked scaffolds (g-PNCOL and gi-PNCOL). An increased expression of pro-inflammatory markers within genipin-crosslinked scaffolds is intriguing, because several in vivo studies demonstrated the anti-inflammatory capacity of genipin molecules [78,79,80,81]. The reason for this discrepancy seems to stem from how the body metabolizes genipin into more biologically active forms. In vivo, some catalases help break down GP into other metabolites that possess anti-inflammatory properties [82] under an acidic environment (≈3–5 pH) [83]. In the in vitro culturing environment, pH is adjusted to ≈ 7–8 for cell survival, which may reduce the anti-inflammatory capacity of genipin. Switching over from the M1 to M2 markers, IL-4 released from the i-PNCOL scaffold significantly increased the anti-inflammatory markers (M2) CCL-18, CD206, IL-10, and CD163 (Figure 9), while the genipin crosslinking lowered anti-inflammatory markers gene expression below detectable levels. The IL-10 expression of macrophages increased 18 ± 0.75 fold, CCL-18 expression increased 9.8 ± 0.5 fold, CD206 expression increased 12 ± 0.5 fold, and CD 163 expression increased 5.5 ± 0.5 fold when they interacted with IL-4 loaded scaffold (i-PNCOL) compared to the scaffold without IL-4 encapsulation. The significant expression of anti-inflammatory markers indicates there was proper activation of the signaling cascade that potentiates M2 differentiation toward resolving the inflammation and promoting the tissue regeneration [84,85,86,87,88,89,90].

This work has some limitations as well. The studies were conducted under in vitro settings, which can be different from the native injured tissue areas. In a complex in vivo environment, the release of IL-4 and immunomodulation capacity of the scaffolds can be different depending on the various factors. Examination of the injectable immunomodulatory tissue scaffold with and without genipin crosslinking in an injured tissue model shall be interesting and exciting to research. However, before running such a comprehensive in vivo study, a systematic in vitro investigation, similar to this study, is a prerequisite to optimizing the parameters used in immunomodulatory scaffold design. 

## 4. Method and Materials

### 4.1. Synthesis of Injectable Immunomodulatory Scaffold 

The injectable immunomodulatory scaffold was synthesized through interspersing interleukin-4 (IL-4) conjugated polycaprolactone (PCL) nanofibers within collagen type-I solution. This nanofibrous yet injectable immunomodulatory scaffold, called i-PNCOL, was synthesized and crosslinked following the major steps explained below and illustrated in Figure 10. 

PCL Nanofiber Fabrication and IL-4 Conjugation: Briefly, the polycaprolactone (PCL) nanofibers were fabricated by dissolving PCL (Mw = 45,000, Sigma-Aldrich, USA) at 16% (*w*/*v*) concentration in an organic solvent mixture (3:1 mixture of chloroform/ methanol) and extruding PCL solution with a rate of 8 mL/h into an electric field created between the syringe tip (0 kV) and collecting plate (20 kV). The electrospun PCL fibers were further dried for 3 days and homogenized using a high-speed homogenizer (Ultra Turrax) to obtain tiny fragments of electrospun fibers. Our prior studies suggested that to prevent the denaturalization of biologics contact with PCL, its hydrophobicity needs to be lowered [56,91]. Thus, PCL nanofibers then were treated with plasma treatment (Harrick Plasma) for 3 min to reduce its hydrophobicity before IL-4 conjugation. IL-4 (Peprotech, Cranbury, NJ, USA) was used as an anti-inflammatory cytokine in this study. Bovine serum albumin (BSA) (Sigma-Aldrich, St. Louis, MO, USA) and heparin (Sigma-Aldrich, St. Louis, MO, USA) were employed to protect IL-4 from denaturalization. IL-4 was first incubated with heparin and then BSA at a ratio of 1:40:2000, respectively, for 15 min at room temperature based on our established protocols [50,51]. Before mixing with the collagen, the IL-4 solution was incubated with functionalized PCL nanofibers for 20 min at room temperature to allow IL-4 to bind to the surface of PCL. Unless otherwise specified, 350 ng/mL of IL-4 was employed in the immunomodulatory scaffold, i-PNCOL.Neutralized Collagen Type-I Solution Preparation: A concentrated (9.6 mg/mL) and acidic (pH~3–4) collagen type-I (Corning, Corning, NY, USA) solution was diluted to 3 mg/mL and further neutralized with chilled 1N NaOH, 10× phosphate buffer saline (PBS), and deionized water according to the vendor’s protocol.Immunomodulatory scaffold, i-PNCOL, Synthesis, and Crosslinking: Then, the IL-4 conjugated-PCL nanofibers were admixed with neutralized collagen type-I solution at 3% (*w*/*v*) concentration to prepare injectable i-PNCOL. The PCL nanofiber concentration (3%) was chosen based on our prior studies [50]. Then, the i-PNCOL was transferred to the incubator (37 °C, 5% CO_2_) for further collagen polymerization for 1 h. Subsequently, either complete cell culture media, digestive inflammatory solution, or PBS were added to the scaffold for assessing the response of the scaffold to the different environments. Pure genipin (Purity > 98%, Challenge Bioproducts, Taiwan) with molecular weight (MW) = 226.2 g/mol was used as a crosslinking agent due to its collagen crosslinking capacity. Genipin crosslinking stock solutions were prepared by dissolving 1.1%( *w*/*v*) genipin powder (C_11_H_14_O_5_) in sterile PBS through mixing overnight at room temperature. Then, the genipin stock solution was filtered using a 0.21 μm pore-size filter to remove possible contaminants. Then, stock solution was diluted to various concentrations of 0.1%, 0.5%, and 1% (*w*/*v*). Then, the scaffolds were cultured with genipin solution in an incubator (37 °C, 5% CO_2_) for 18 h before further characterizations. On characterization day, the solution was aspirated, and the scaffolds were washed vigorously 3 times with sterile PBS to remove residual genipin solution. The genipin crosslinked scaffolds were referred to as gi-PNCOL throughout the study (Figure 10).

### 4.2. Degradation and Cytokine Retention Capacity of Scaffold in Digestive Inflammatory Environment 

Simulating Digestive Inflammatory Environment: Upon injury, innate immune cells express a high amount of degrading enzymes, especially matrix metalloproteinase-1 (MMP-1) to degrade the tissue debris [92]. Thus, to mimic the digestive inflammatory environment, lyophilized MMP-1 (interstitial collagenase type-1) (Gibco, USA) was utilized as a degrading enzyme after dissolving it in 1mM Ca^+2^ supplemented PBS with a concentration of 1U/mL. 

#### 4.2.1. Measuring Scaffold Degradation in the Digestive Inflammatory Environment 

The degree of scaffold degradation under the digestive inflammatory environment (DIE) was measured through tracking collagen degradation, since MMP-1 cleaves the triple-helical α-chain of collagen, which further leads to the premature degradation of collagen-based tissue scaffolds [93]. The scaffolds were created and cultured in DIE, and the collagen degradation was assessed using the Sircol Assay (Biocolor, Carrickfergus, UK). In brief, the media was collected and replaced with fresh media on days 1, 5, 9, and 14 for analysis. On characterization day, the collected media was mixed with Sircol dye under a moderate agitation condition for 30 min at room temperature and centrifuged. Following centrifugation, the collagen pellet was washed in cold acid-salt wash reagent and suspended in alkali reagent. Then, the solution was transferred to a 96-well plate, and the absorbance was read at 555 nm using a microplate reader (SOFTmax Pro, San Jose, CA, USA). The amount of degraded collagen was calculated based on a standard curve generated with a different known amount of collagen type-I (Corning, Corning, NY, USA). The data were normalized to the amount of collagen initially incorporated within the scaffolds. 

#### 4.2.2. Measuring Kinetic Release of Encapsulated Cytokine in Digestive Inflammatory Environment

The short-term and long-term cytokine retention capacity of scaffolds in DIE and the role of genipin crosslinking to improve the retention were studied through assessing the kinetic release of a model cytokine over 4 days (short-term) and 21 days (long term) within a degrading environment. Lysozyme was chosen as a model cytokine in the release study because it contains a similar isoelectric point (Ip 9–10) [94], molecular weight (MW 14.5 kDa), and molar extinction coefficient (εk 38,000 cm^−1^M^−1^) to IL-4 [95]. This polypeptide has been extensively used to understand encapsulated biologic diffusion from scaffolds and a scaffold’s potential to effectively load and release biologics [96,97]. Lysozyme (Sigma-Aldrich, St. Louis, MO, USA) was covalently labeled with Alexa Fluor 350 dye (Molecular Probes) and purified by size exclusion chromatography using Sephadex G-25 resin column (Sigma-Aldrich, St. Louis, MO, USA). Briefly, the Alexa-Fluor-350 tagged lysozyme was incorporated into scaffold solutions at a final concentration of 20 μg/mL. First, the role of DIE on the retention capacity of the scaffold was investigated through culturing scaffolds in DIE or PBS (control group). Then, the role of genipin crosslinking was studied through culturing scaffolds to DIE before and after crosslinking. Briefly, on characterization day, supernatant from each sample was aspirated completely and replaced with an equivalent volume of DIE or PBS. Then, the collected samples were transferred into 96-well UV-Plates (Thermo Fisher, Waltham, MA, USA) and analyzed for fluorescence intensity using a microplate fluorometer (Wallac 1420) at 346/442 nm excitation/emission wavelengths. The amount of model cytokine released at each characterization day was determined from the standard curve generated by plotting the fluorescence intensities of known concentrations of the dye-labeled lysozyme. The release profile of lysozyme from the scaffolds was determined at 1, 2, 3, and 4 h post-incubation to study the initial burst, followed by 1, 2, 4, 6, 10, 14, 17, and 21 days to study protein release over a longer period of time. The amount of released protein was normalized to the initial amount of protein loaded per sample. Four samples (*n* = 4) were used for experimental and control groups.

### 4.3. Genipin-Mediated Cytotoxicity and Morphological Changes in Scaffold

#### 4.3.1. Identifying Degree of Crosslinking Following Genipin 

The ninhydrin assay was used to understand whether genipin crosslinks the scaffolds and genipin concentration affects the degree of crosslinking through measuring the free amine groups before and after crosslinking. On characterization day, the scaffolds were taken out from cultured media, snap-frozen, and immediately weighed. Then, the samples were incubated with a ninhydrin solution (Sigma-Aldrich, St. Louis, MO, USA) created by following a well-established protocol [98]. Following 10 min incubation at room temperature, the solutions were mixed on a vortex mixer; then, they were covered and heated to 100 °C in 10 M HCl for 15 min. The tubes were allowed to cool for an additional 10 min and neutralized to a pH of 7 with 10 M NaOH. The contents of the tubes were mixed thoroughly, and the absorption at 404 nm of the solutions was measured using a spectrophotometer (SOFTmax Pro, San Jose, CA, USA). A calibration curve was created by measuring the adsorption of different concentrations of glycine from 0 to1.6 mg/mL in deionized water (0, 0.4, 0.8, 1.2, and 1.6 mg/mL) and ninhydrin solution at 404 nm with a spectrophotometer (SOFTmax Pro, San Jose, CA, USA).

#### 4.3.2. Measuring Cell Toxicity upon Genipin Crosslinking

C2C12 cells (ATCC, Manassas, VA, USA) were used in the toxicity study. The cells were cultured in DMEM supplemented with 10% fetal bovine serum (FBS) and 1% penicillin/streptomycin (Thermo Fisher, Waltham, MA, USA). Upon confluency, PCL nanofibers and neutralized collagen scaffolds were prepared as described previously and admixed with C2C12 cells with 1 × 10^6^ cells/mL seeding density. The 0% (control), 0.5%, and 1% (*w*/*v*) genipin crosslinking solutions were also prepared using cell culture media as described in Section 2.1. Then, cell-encapsulated scaffolds were incubated within the genipin solutions for 18 h. Following the incubation, the scaffolds were washed twice with PBS, and the media was replaced with complete cell culture media for further seven-day incubation. The DNA quantification was performed on days 1, 3, and 7 to indirectly determine the total number of cells within scaffolds using PicoGreen dsDNA kit (Thermo Fisher, Waltham, MA, USA). Briefly, on characterization day, the cells were liberated from the collagen using snap-freezing followed by mechanical disruption with a homogenizing pestle. Then, the crushed scaffolds were resuspended in lysis buffer (50 mM Tris HCl, 1 mM CaCl_2_, 400 µg/mL proteinase K at pH = 8) and incubated at 55 °C overnight. The lysate was diluted 1:10 in TE buffer and mixed with 1:200 dilution of PicoGreen dye in a 1:1 ratio. Following incubation at room temperature for 5 min, the samples were moved to a microplate fluorometer (Wallac1420) to measure their fluorescence intensities at 480/520 nm excitation/emission wavelengths. Then, the total amount of DNA was determined using a standard curve generated with varying amount of DNA in ng and their corresponding fluorescence values. 

#### 4.3.3. Assessing Changes in Scaffold Morphology upon Genipin Crosslinking

The changes in scaffolds’ morphology, fiber diameter, and porosity following the genipin crosslinking were examined using Scanning Electron Microscope (SEM). The scaffolds before and after genipin crosslinking were fixed overnight with 4% paraformaldehyde; then, the scaffolds were sequentially dehydrated by incubating them for 15 min each in a series of ethanol/water gradients followed by hexamethyldisilazane/ethanol gradients ranging from 30% to 100%. Then, the scaffolds were air-dried overnight, sputter-coated with gold, and visualized under SEM to observe the morphological and structural changes upon genipin crosslinking and how these changes were dependent on genipin crosslinking concentration.

#### 4.3.4. Measuring Viscoelastic Properties of the Injectable Scaffold upon Genipin Crosslinking

To determine the role of genipin crosslinking on mechanical properties of the scaffolds, rheological studies (*n* = 6) were carried out at 25 °C using a TA Instruments Discovery Hybrid Rheometer (DHR-3) with an 8 mm diameter stainless steel parallel plate geometry and 0.5 mm gap. To investigate the relationship between molecular structure and viscoelastic behavior, experiments were conducted in the region where viscoelastic properties were independent of stress. The linear viscoelastic region is where storage modulus, G’, and loss modulus, G”, are independent of stress, and the sample structure is undisturbed [99]. The linear viscoelastic region was determined by conducting strain sweeps on the scaffolds crosslinked with genipin at three different concentrations. The storage modulus (G’) was determined by performing frequency sweep tests in the range of 0.01−1 Hz at fixed stress within the linear viscoelastic region of the scaffolds (5–20 Pa) [100].

### 4.4. Assessing Immunomodulatory Capacity of Scaffold 

Immunomodulatory capacity of the scaffold with and without genipin crosslinking was assesed through culturing the monocyte-derived macrophages with the scaffold. Figure 11 illustrates the major experimental groups and the outline of immunomodulatory assestment of the scaffold.

#### 4.4.1. Monocytic Cell Expansion and Differentiation to Macrophage

The human pro-monocytic cell line U937 (ATCC, Manassas, VA, USA) was maintained in suspension culture in Roswell Park Memorial Institute (RPMI) 1640 medium (ATCC, Manassas, VA, USA), supplemented with 10% (*v*/*v*) heat-inactivated fetal bovine serum (FBS, company), 2 mM L-glutamine, 10 mM HEPES, 1 mM sodium pyruvate, 4500 mg/L glucose, and 1500 mg/L sodium bicarbonate and kept at the incubator (37 °C, 5% CO_2_) for expansion. The medium was changed every 2–3 days to maintain cell densities of 1.0 × 10^6^ cells/mL. Then, U937 cells were differentiated into macrophages (M0) through incubating with 100 ng/mL of phorbol 12-myristate 13-acetate (PMA) (Sigma-Aldrich, St. Louis, MO, USA) in complete media for 24 h. The following day, the media was aspirated, and fresh complete media was added. Cells were allowed to attach and acclimate for 36 h before characterization. 

#### 4.4.2. Response of Macrophage Polarization to Immunomodulatory Scaffolds 

The effect of IL-4 released from the immunomodulatory scaffold on macrophage (M0) polarization was assessed through the mRNA expression of pro- and anti-inflammatory markers following the incubation of macrophages (M0) with IL-4-encapsulated, IL-encapsulated and genipin crosslinked, and control scaffolds (PNCOL). The role of genipin crosslinking on the immunomodulatory capacity of the scaffold was further investigated. Figure 2 demonstrates the study design for macrophage polarization using immunomodulatory scaffolds i-PNCOL and gi-PNCOL.

After PMA treatment and subsequent overnight incubation (as described above), the cells were allowed to adhere to the bottom of 12-well transwell plates at a density of 1 × 10^6^ cells/well. The immunomodulatory scaffold (i-PNOCL) or genipin-crosslinked immunomodulatory scaffold (gi-PNCOL) were synthesized (as described in Section 4.1) and transferred to the top chamber of transwell culture inserts (Ø = 12 mm, pore size 3 μm, Corning, Corning, NY, USA). The wells were filled with fresh RPMI media (1.5 mL) and placed in the incubator for 4 days.

Following incubation, the polarization of macrophages exposed to IL-4 released from i-PNCOL and gi-PNCOL was studied by performing expression analysis of pro-and anti-inflammatory genes through quantitative real-time polymerase chain reaction (qRT-PCR). Briefly, cells were collected, and RNA was extracted using TRIzol reagent (Thermo Fisher, Waltham, MA, USA), and reverse transcription was performed using the Omniscript RT kit (Qiagen, US) per the manufacturer’s instructions. Quantitative real-time PCR was performed using SYBR Green PCR master mix (Thermo Fisher, Waltham, MA, USA) to detect the expression of pro-inflammatory genes (MMP-3, MMP-9, TNF-α, COX-2) and anti-inflammatory genes (CCL-18, CD206, CD163, IL-10). All expression assays were normalized to glyceraldehyde-3-phosphate dehydrogenase (GAPDH). Primer sequences were obtained from published literature as listed in Table 1 and purchased from Integrated DNA Technologies. PCR was performed using iCycler iQ detection system (Biorad, California, USA) with thermocycling performed for 40 cycles. Data were analyzed for fold differences in gene expression with respect to control samples using the ΔΔ*C*t method.

### 4.5. Statistical Analysis

Statistical analysis was conducted through Minitab. Statistical analysis was performed using Student’s *t*-test and Oneway ANOVA. All values are reported as the mean and ± the standard deviation of the mean. *p* < 0.05 was considered to be statistically different.

## 5. Conclusions

Understanding the immunomodulatory capacity of anti-inflammatory cytokine loaded tissue scaffolds and how they behave under the digestive inflammatory environment are indispensable for improving the therapeutic strategies for different musculoskeletal tissue injuries. In this study, we demonstrated that the digestive inflammatory environment created premature degradation of the immunomodulatory scaffold, which further caused the burst release of loaded IL-4 from the scaffold. We eliminated the deleterious effect of the digestive inflammatory environment using genipin crosslinking. The comprehensive structural, mechanical, and biological analysis conducted on genipin-crosslinked immunomodulatory scaffolds suggested that genipin-crosslinking should be utilized with caution. The genipin-crosslinking can be beneficial in creating a sustained release of IL-4 from the scaffolds, but the genipin-crosslinked collagen scaffold may not be the best option as an immunomodulatory scaffold. Genipin crosslinking promoted the pro-inflammatory markers expressions while lowering anti-inflammatory markers expressions below the detectable level. On the other hand, IL-4 released from the immunomodulatory scaffold (i-PNCOL) promoted anti-inflammatory lineage commitment of naïve macrophages.

## Figures and Tables

**Figure 1 ijms-22-01134-f001:**
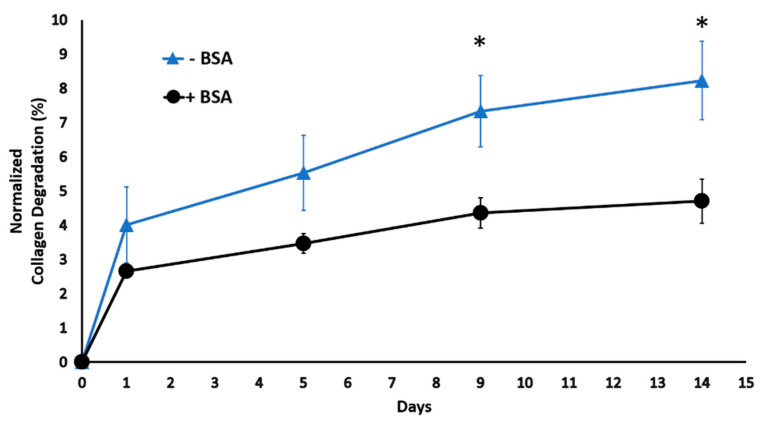
The kinetic degradation of the scaffold with and without BSA under a digestive inflammatory environment. * *p* < 0.05.

**Figure 2 ijms-22-01134-f002:**
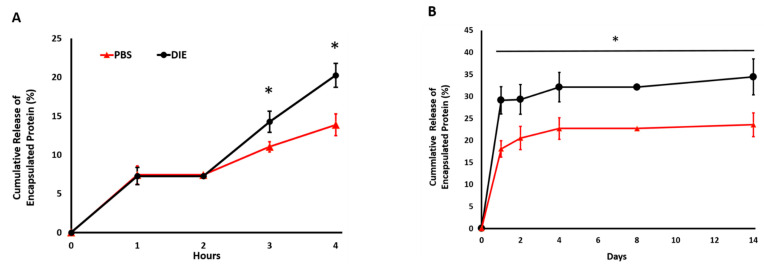
The role of digestive inflammatory environment (DIE) in model protein release profile from injectable scaffolds. (**A**) Model protein release profile within injectable scaffolds (*n* = 4) cultured in phosphate buffer saline (PBS) and DIE during time = 0 to 4 h corresponding to the burst release phase and (**B**) over 14 days. * *p* < 0.05.

**Figure 3 ijms-22-01134-f003:**
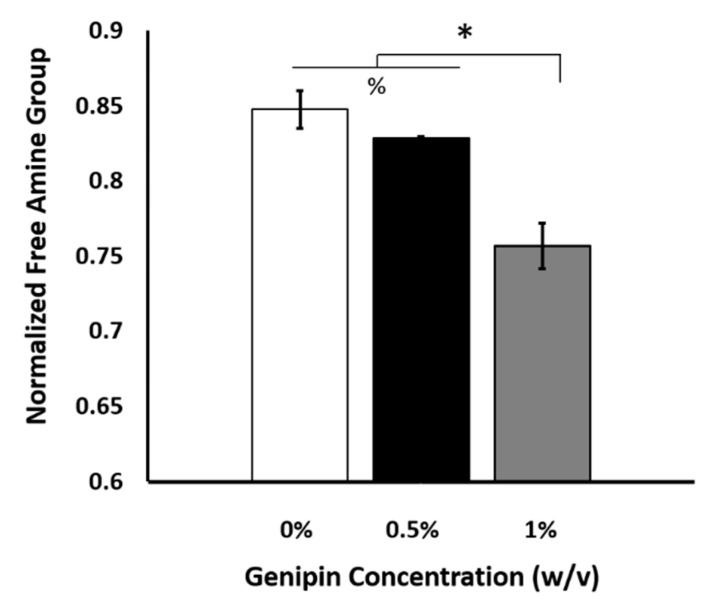
The changes in the scaffolds’ degree of crosslinking with the increased genipin concentration. (*) indicates a significant difference between the experimental groups, *p* < 0.05 while (%) indicates a significant difference between the experimental groups, *p* < 0.01.

**Figure 4 ijms-22-01134-f004:**
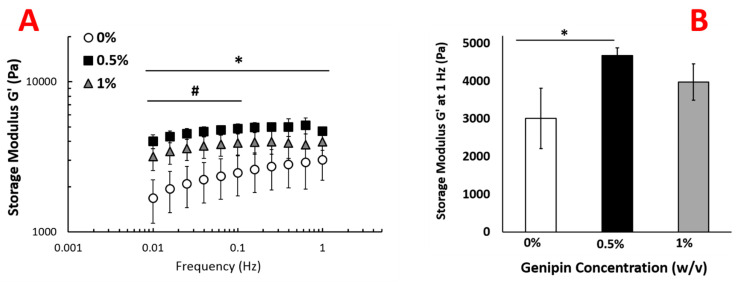
Rheological analysis of genipin crosslinked scaffolds (**A**) The changes in the storage modulus as a function of frequency for scaffolds crosslinked with various genipin concentration (0%, 0.5%, and 1% (*w*/*v*)). (**B**) Storage modulus at 1 Hz as a function of genipin concentration. Six samples were used for each group (*n* = 6). (#) denotes a significant difference between frequencies, while (*) indicates a significant difference between sample groups, *p* < 0.05.

**Figure 5 ijms-22-01134-f005:**
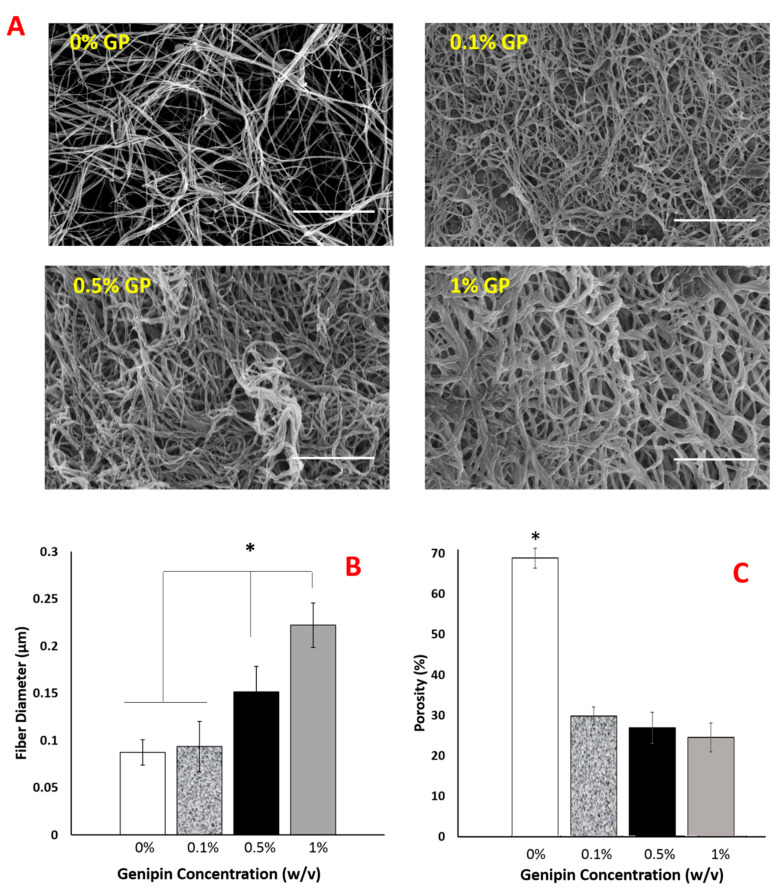
The morphological changes within the scaffolds crosslinked with various genipin concentrations. (**A**) SEM images of scaffolds show increased fiber thickness with increased genipin concentration. Scale bar is 5 µm for all SEM images. (**B**) The changes in fiber diameter and (**C**) porosity as a function of genipin concentration. (*) indicates a significant difference between sample groups, *p* < 0.05.

**Figure 6 ijms-22-01134-f006:**
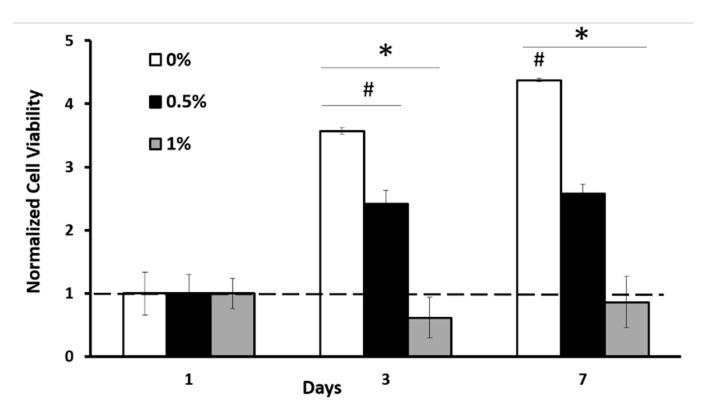
Cytotoxicity effect of genipin crosslinking with different genipin concentrations. # denotes a significant difference between time points, while (*) indicates a significant difference between sample groups, *p* < 0.05.

**Figure 7 ijms-22-01134-f007:**
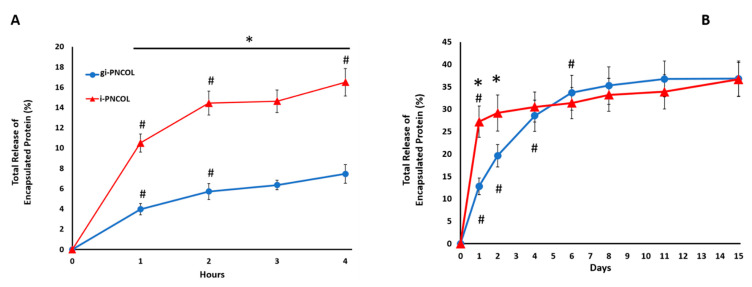
Impact of genipin crosslinking on model protein release from scaffolds under digestive inflammatory environment. (**A**) The release profile during the first 4 h corresponding to the burst release phase and (**B**) over 15 days. # denotes a significant difference between the time point points. * indicates a significant difference between groups, *p* < 0.05.

**Figure 8 ijms-22-01134-f008:**
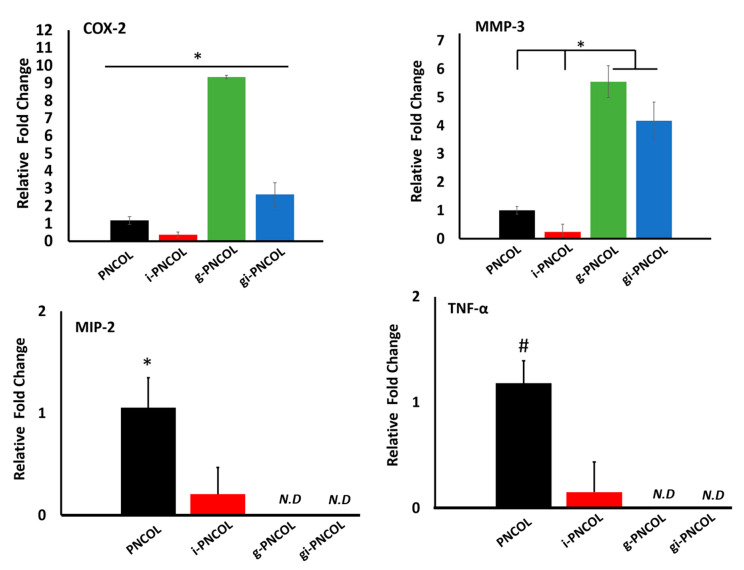
The relative expression of pro-inflammatory (M1) Genes. Star sign (*) indicates a significant difference between sample groups (*p* < 0.05). The pound sign (#) indicates a significant difference between sample groups (*p* < 0.1). N.D indicates Not Detected gene expressions.

**Figure 9 ijms-22-01134-f009:**
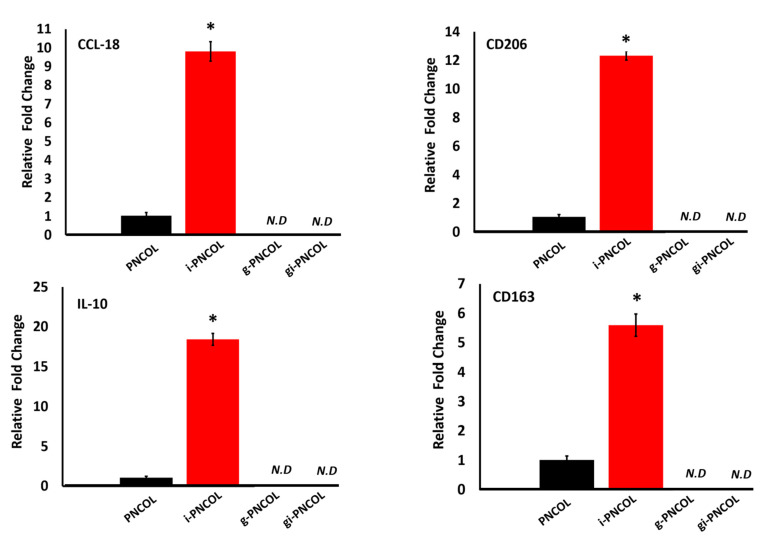
The relative expression of anti-inflammatory (M2) genes. Star sign (*) indicates a significant difference between sample groups. Student’s paired t-test was used, *p* < 0.05. N.D indicates Not Detected gene expressions.

**Figure 10 ijms-22-01134-f010:**
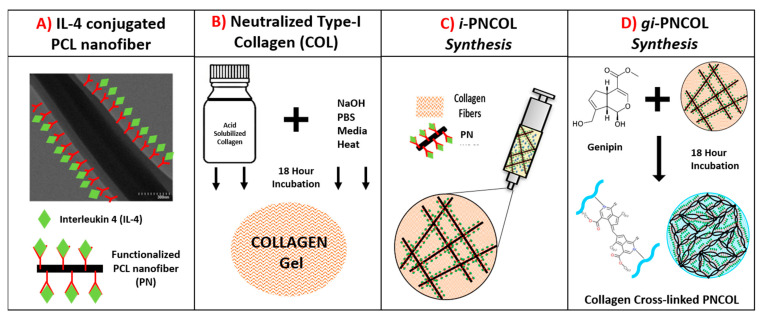
The major steps to synthesize injectable immunomodulatory scaffold, i-PNCOL, and genipin-crosslinked scaffold, gi-PNCOL. (**A**) Interleukin-4 (IL-4) conjugation on functionalized polycaprolactone (PCL) nanofibers, (**B**) Neutralizing collagen solution, (**C**) Incorporation of IL-4 conjugated PCL nanofibers in collagen to create i-PNCOL, (**D**) crosslinking i-PNCOL using genipin to create gi-PNCOL.

**Figure 11 ijms-22-01134-f011:**
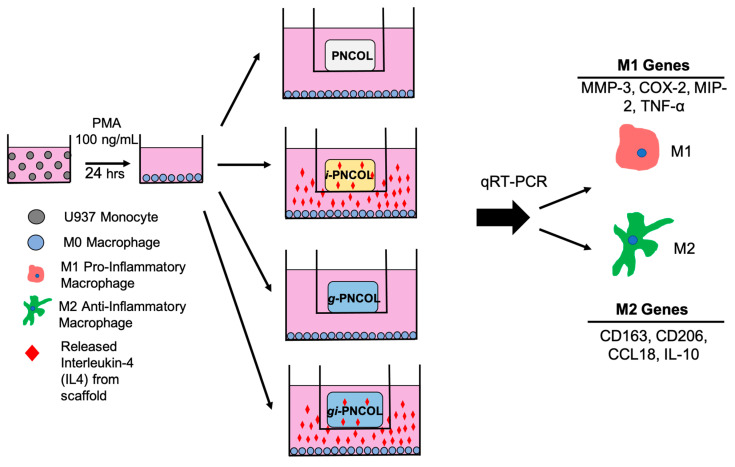
Study design for macrophage polarization upon IL-4 release from the genipin crosslinked and uncrosslinked scaffold.

**Table 1 ijms-22-01134-t001:** Forward and reverse primers for real-time PCR.

Gene	Forward Primer	Reverse Primer	Ref
MMP-3	5′-CAAGGAGGCAGGCAAGACAGC-3′	5′-GCCACGCACAGCAACAGTAGG-3′	[101]
COX-2	5′-CGGTGTTGAGCAGTTTTCTCC-3′	5′-AAGTGCGATTGTACCCGGAC-3′	[102]
MIP-2	5′-CGCCCAAACCGAAGTCAT-3′	5′-GATTTGCCATTTTTCAGCATCTTT-3′	[103]
TNF-α	5′-AGAGGGAAGAGTTCCCCAGGGAC-3′	5′-TGAGTCGGTCACCCTTCTCCAG-3′	[104]
CD163	5′-TCTGTTGGCCATTTTCGTCG-3′	5′-TGGTGGACTAAGTTCTCTCCTCTTGA-3′	[105]
CCL18	5′-AAGAGCTCTGCTGCCTCGTCTA-3′	5′-CCCTCAGGCATTCAGCTTCA-3′	[106]
IL-10	5′-CCTGTGAAAACAAGAGCAAGGC-3′	5′-TCACTCATGGCTTTGTAGATGCC-3′	[107]
CD206	5′- CTACAAGGGATCGGGTTTATGGA-3′	5′- TTGGCATTGCCTAGTAGCGTA-3′	[108]
GAPDH	5′-AGAAGGCTGGGGCTGATTTG-3′	5′-AGGGCCCATCCACAGTCTTC-3′	[109]

## Data Availability

The data is presented based on MDPI Research Data Policies which can be accessed at https://www.mdpi.com/ethics.

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
