# Peer review of "Impact of Digestive Inflammatory Environment and Genipin Crosslinking on Immunomodulatory Capacity of Injectable Musculoskeletal Tissue Scaffold"

_ijms, 2021, doi:10.3390/ijms22031134_

Round 1

Reviewer 1 Report

     This paper is a study of Immunomodulatory capability of genipin crosslinked PCL-collagen injectable hydrogels in the presence of digestive inflammatory environment. To control local inflammation and ensure tissue regeneration, immunomodulatory scaffolds are developed to release anti-inflammatory cytokines. However, I do not think that the inflammatory environment was not yet well-studied if they have any effects on the cytokine release, scaffold degradation, and the immunomodulatory capability. The authors chose collagen as the material of the scaffold because of their relatively low immunogenicity and injectability. Here the genipin crosslinking is a promising method to control the burst release of encapsulated cytokines, but the effect of genipin crosslinking to immunomodulation is also not yet well studied. In this research, the authors first compare the release profile of IL-4 in PBS to digestive inflammatory environment. Then, the degree of crosslinking, mechanical properties, and morphology of hydrogels are investigated. After that, cell cytotoxicity, IL-4 release profile among crosslinked and non-crosslinked hydrogels, and lastly macrophage polarization is measured. However, some of the figures need adjustment and some of the methods did not inform the population size or sample size. Taken together, major revision should be made before paper re-submission.

1) Results (Page 5, Figure 4)                                                                    What does the symbol “*” and “#” refers to in Figure 4A?

2) Results (Page 6, line 9)                                                                              “The dramatic change in fiber diameter was observed for 1% genipin crosslinked-PNCOL scaffolds” This is an important statement. Symbol of significant difference should be included in Figure 5B.

3) Methods and Materials (Page 14, line 25)                                                  How long was IL-4 solution incubated with functionalized PCL nanofibers?

4) Materials and Methods (Page 15, line 15)                                              “1.1 (w/v) genipin powder” should be changed into “1.1 %(w/v) genipin powder”

5) Materials and Methods (Page 16, line 29)                                              Who is the manufacturer of ninhydrin assay?

6) Materials and Methods (Page 18, Figure 11)                                                The incubation time of PMA should be changed from 18 hrs to 24 hrs.              The g-PNCOL conditions should also be inserted in this figure. 

7) Is the time profile of IL-4 release optimized for the purpose of this study?

8) Figure 5: the fiber diameters of scaffolds are changed by the genipin concentration. The change must affect the biological function of cells. This point should be discussed.

9) Among the chemical crosslinking agents, the reason why genipin was used in this study should be described. What is the advantage or disadvantage of genipin comparing with other agents?

10) Figure 9: the phenotype change of macrophages is evaluated by the CD antigens. How about the cytokine secretion of M2 macrophages?

Reviewer 2 Report

Overall the paper is lenghty and difficult to read. The article reads like a masters student thesis copied into a paper format.

Author Response

The authors thank the reviewer for the comment and sorry for not being concise enough. This study is a comprehensive study with multiple facets so it was very hard to make short without compromising the important detail about the study. The authors excluded as many details as possible yet still, the authors also agree that it is relatively a lengthy manuscript.

Reviewer 3 Report

In this paper, the authors investigated the immunomodulatory capacity of anti-inflammatory cytokine loaded tissue scaffolds and how they behave under the digestive inflammatory environment are indispensable for improving the therapeutic strategies for different musculoskeletal tissue injuries. The article is interesting, limited by only in vitro studies as no in vivo studies had been carried out. A few comments to address:

1. Line 474—Authors are advised to mention how many milliliters of NaOH and 10X phosphate buffer saline (PBS), and deionized water were used to prepare the Neutralized Collagen Type-I Solution. The utilization of NaOH for the hydrogel preparation may lead to deproteinization and it may affect the function of Interleukin-4. Please justify it clearly.

2. L 105, What is the reason for the sudden degradation of both groups of scaffolds on day one? Give a brief explanation.

3. L 108-110, Authors emphasize the importance of BSA in maintaining the structural stability of the scaffolds in the digestive environment. In case, if the scaffold is to be used for the tissue engineering application, BSA will be consumed by the cells and the scaffold will lose its structural stability. How does this observation relate to the claim made in this article "Bovine Serum Albumin Diminishes Scaffold for Degradation in Digestive Inflammatory Environment"?

4. L 119, Figure 4A should be corrected as Figure 2A.

5. L 181, In the Figure, caption for C is missing and or line 184 should be corrected as (B&C) Porosity and fiber diameter data from the analysis of SEM images.

6. In figure 5, only the SEM image 0.1% GP has a scale bar and the other three images don’t have the scale bar. Keep the scale bar on every image as they were taken at different magnifications.

7. In Figure 4, the figure caption doesn’t clearly mention which is for A and B. 

8. In figure 5, the SEM images of scaffolds with 1% genipin concentration shows increased fiber thickness. But mechanical properties like the storage modulus were observed to be less when compared to the 0.5%. Why?

9. In figure 3, rewrite the figure caption clearly.

10. L 88. the objectives of the study were repeated again in line 285 in the discussion part. It should be removed.

11. L 313. Explain clearly how the degradation of scaffold due to inflammation may affect the release profile of encapsulated cytokines within the scaffolds with proper justifications.

12. L 314. How is the information on the digestive inflammatory environment affecting the kinetic release profile of anti-inflammatory cytokines help in identifying the optimal cytokine?

13. L 316. How the kinetic release studies in the digestive inflammatory environment support in identifying the optimal cytokine level and how it can be incorporated within the scaffold to modulate the inflammation in an injury in an in vivo environment. Provide comprehensive information with reference.

14. L 119,120. Why were the scaffolds incubated in two different medium such as DIE and PBS showing the same trends in the first and second hours with total protein release stable at 7.3±1.2%? L 122. In contrast, the release percentage from scaffolds cultured in PBS demonstrated the same trend for the first four hours, and the release rate was significantly less than scaffolds in DIE at the 3rd and 4th hour. Why?

15. There are a lot of typos throughout the article, a few mentioned above. Do go through the entire article for identifying and correcting them.

Round 2

Reviewer 1 Report

The authors added some information about the comments pointed out, and the most of concerns were resolved. However, the caption of figure 4A should be revised for readers’ better understanding. Minor revision should be made before paper re-submission.

  • In figure 4A there is the relationship between storage modulus and frequency. Therefore, the caption “# denotes a significant difference between time points” should be referring to the varied frequencies instead of time points.

Author Response

The authors thank the reviewer for the comment. The authors agree with the reviwer and revised the caption of Figure 4 for better understanding. 

In the revised manuscript Figure 4 caption (Page 6 line 186) changed to 

Figure 4. Rheological analysis of genipin crosslinked-scaffolds  (A) The changes in the storage modulus as a function of frequency for scaffolds crosslinked with various genipin concentration (0%, 0.5%, and 1% (w/v)) (B) Storage modulus at 1 Hz as a function of genipin concentration. Six samples were used for each group (n=6). (#) denotes a significant difference between frequencies while (*) indicates a significant difference between sample groups, p < 0.05.  

Reviewer 2 Report

Thanks for the update

Author Response

Thank you for your time and input. 

Reviewer 3 Report

All comments are satisfactorily addressed by the authors.

Author Response

Thank you for your time and input.